# Investigation of Aircraft Conflict Resolution Trajectories under Uncertainties

**DOI:** 10.3390/s24185877

**Published:** 2024-09-10

**Authors:** Anrieta Dudoit, Vytautas Rimša, Marijonas Bogdevičius

**Affiliations:** 1Department of Aviation Technologies, Vilnius Gediminas Technical University (VILNIUSTECH), LT-10223 Vilnius, Lithuania; vytautas.rimsa@vilniustech.lt; 2Department of Mobile Machinery and Railway Transport, Vilnius Gediminas Technical University (VILNIUSTECH), LT-10223 Vilnius, Lithuania; marijonas.bogdevicius@vilniustech.lt

**Keywords:** aircraft conflict, conflict resolution, the Dubins method trajectory, dynamic conflict solution, uncertainty, wind

## Abstract

As air traffic intensity increases and stochastic uncertainties, such as wind direction and speed, continue to impact air traffic controllers’ workload significantly, airlines are increasingly pressured to reduce costs by flying via straighter/more direct trajectories. Due to these changes, it is important to search for new means/solutions for aircraft conflict resolution to ensure the required level of safety and rational flight trajectory. Such a solution could be the implementation of Dubin’s method of flight trajectories. This paper aims to propose and deeply analyze a new mathematical model for two-aircraft conflict resolution where the Dubins method is applied in a dynamic conflict scenario. In this model, at a certain moment, the flight trajectory of one aircraft follows a path similar to a moving circle’s tangential line. Upon that, the conflict detection and resolution (CDR) model considers wind uncertainty. The proposed CDR method could be applied when uncertainty such as wind direction and speed are inconstant (stochastic) throughout the simulation.

## 1. Introduction

The safe movement of aircraft (in the air) is the aviation industry’s top priority, particularly in light of the predicted increase in the number of aircraft flights [1,2]. After 2025, flight growth is expected to average 1.5% per year in the base forecast, owing to the greater uncertainties within the 7-year horizon (higher inflation, pressure on oil prices, and environmental concerns) [3]. A higher density of flights requires a more precise investigation of aircraft conflicts with respect to flight uncertainties and safety requirements.

Wind is one of the primary sources of uncertainties in aircraft operations, potentially disrupting all scheduled aircraft flight operations [4,5]. More accurate planning, considering wind effects, could improve aircraft flight paths and enable the safe and rational resolution of conflict situations [6,7]. For this reason, wind uncertainties are important because they can both positively and negatively impact aircraft conflict situations resolution and rational trajectories optimization, resulting in differences in the national and overall European air traffic network throughput rate/functioning [2,8,9,10]. 

The Air Traffic Management (ATM) system (from a ground-based perspective) has a set of decision support system (DSS) functions designed to help assist traffic controllers (ATCOs) in detecting and resolving aircraft conflict situations. One of the functions of DSS is trajectory prediction, which is used to provide guidance on forecasted aircraft future positions [11]. 

Some DSS functions for conflict detection and resolution (CDR) on a horizontal plane include MTCD (Medium-term Conflict Detection), TCT (Tactical Controller Tool) and the ground-based safety net—STCA (Short-term Conflict Alert). The final DSS or last resort (in the vertical plane) is aircraft-based safety net, named ACAS/TCAS (Airborne/Traffic Collision Avoidance System) [11] as presented in Figure 1.

In the nowadays aviation industry, the most typical methods for solving conflicts are provided in the documents [12,13,14] and the use of deterministic and stochastic methods for conflict resolution have been explored in more detail in the research papers [15,16,17,18,19,20] and the most typical methods are provided below:*SLOT time*—represents a specific, scheduled time allocated to an airline for either takeoff or landing at a congested airport. *Advantages:* This time slot is a critical component of air traffic management strategies aimed at controlling and optimizing the flow of aircraft movements in and out of airports facing high traffic volumes. *Disadvantages:* This method application may induce congestion of the airport [21].*Level change*—this solution is typically used for resolution of the conflicting aircraft in level flight. *Advantages:* This is the quickest mean of aircraft conflict resolution. *Disadvantages:* This method may trigger ‘domino effect’ with other aircraft flying at lower or higher flight levels [22] and moreover not all horizontal airspace resources are used [23].*Speed control*—this method is mostly suitable for solving medium-term conflicts (as the instruction takes time to “produce” separation) and for maintaining already achieved separation. *Disadvantages:* A major limiting factor is the typical cruising speed [24,25,26].*Vertical speed control*—this technique is used in situations where an aircraft needs to safely cross another aircraft’s level. *Advantages:* If properly used, it provides safe and efficient flow of traffic. *Disadvantages:* Also, while descent rates are usually achievable, climbing at a specified vertical speed may be outside the aircraft capability and therefore the restriction should be coordinated with the flight crew [24,25].*Heading change*—this is a universal method. *Advantages:* This method may solve any conflict unless additional factors (such as airspace restrictions or adverse weather) limit the available headings [27]. *Disadvantages:* This benefit comes at a cost since vectoring usually extends the distance flown (and hence, delays the flight). This effect may sometimes be mitigated by providing a direct routing after the conflict has been solved [28,29,30,31]. Because of the fact that this method is universal and perspective therefore its mathematical modelling and theoretical application will be further analyzed in this article.

Some methods provide more flexibility for en-route trajectory planning and greater responsibility for self-separation [32,33,34]. However, conflicts may arise between an unmanned aerial vehicle (aircraft piloted by remote control or onboard computers) and manned aerial vehicles (aircraft piloted directly by humans) [35,36]. An exhaustive review of current CDR methods for manned and unmanned aviation presents a taxonomy that categorizes algorithms in terms of their approach to conflict-avoidance planning, surveillance, control, and trajectory propagation [15,37,38].

Machine learning methods [7] could potentially adapt to the detrimental emergent behaviors from multi-actor conflicts and the knock-on effects of successive conflict resolution maneuvers that occur as traffic densities increase [39,40].

The literature review revealed that stochastic uncertainties play a significant role in ATCOs workload, together with pressure from airlines to reduce costs by flying via straighter/more direct trajectories [41,42,43,44]. This highlights the value of offering a new model for an aircraft conflict resolution advisor that ensures safe, economical, and rational management of aircraft conflict situations [45,46,47,48].

This paper aims to propose and deeply analyze a new mathematical model for two-aircraft conflict resolution where the Dubins method is applied in a dynamic conflict scenario. In this model, at a certain moment, the flight trajectory of one aircraft follows a path similar to a moving circle’s tangential line. Upon that the CDR model, considers wind uncertainty. The proposed CDR method can be applied when uncertainty such as wind direction and speed are variable (stochastic) throughout the simulation. When a collision between two aircraft is detected, the starting point of the potential aircraft collision is determined as the center of the circle, with the circle’s radius being greater than 5 nm. One aircraft moves in a circle, and the other in a straight line. As the aircraft continues to fly, the center of the circle moves together with the aircraft flying in the the circular path, with its speed and direction of movement of the center of the circle aimed toward the aircraft’s final simulation point. At each moment in time, the aircraft flight coordinates are determined by numerical method, by integrating the variations in the coordinate vector equations using the trapezoidal method. The integration time step is chosen according to the desired accuracy of the solution.

At every moment in time, the direction of motion of the aircraft flying along the circle’s perimeter is close to the tangent of the circle. At each moment in time, the arc of the moving circle is divided into five-node discrete elements, allowing for a more accurate determination of the radius of the curvature and the tangent and normal to the circle of the aircraft trajectory.

The literature review has shown that while methods exist for addressing aircraft conflict resolution, they are ineffective in dealing with the wind impact. Thus, we propose a mathematical model suitable for safe and rational aircraft conflict resolution on a horizontal plane. This model considers the analyzed possible initial angles between two aircraft flight trajectories (green area), angles not accepted for investigation (red area) and angles not applicable for investigation due to International Civil Aviation Organization (ICAO) and European Union Aviation Safety Agency (EASA) cruising level rules (grey area), which require aircraft flying at the same flight level to fly in the same direction [49]. Hence, opposite traffic is excluded from the scope of analysis, and the analyzed areas are shown in Figure 2a,b.

## 2. Simulation Assumptions

Hence, the analytical investigation model was established as follows (Figure 3).

The aim is to increase the distance between aircraft flying on flight path $A_{1}A_{2}$ (mode 1–4) to ensure a safe horizontal separation of the 5 nm between the two aircraft flight trajectories $A_{1}A_{2}$ and $B_{1}B_{2}$ at every time step, with Δt = 0.01 s.

The initial radii of the two circles centered $O_{s}$ and $O_{1}$ were assumed to be 5 nm. However, to ensure the safety condition, i.e., maintaining a minimum horizontal distance of not less than 5 nm between the flight paths $A_{1}A_{2}$ and $B_{1}B_{2}$ of the two aircraft (for initial angles between such flight trajectories being 35°, 45° and 55°) at each point in time, the radius of the $r_{{Os}_{o}}$ has to be incremented by the $\Delta_{r_{OS}}$ step until the $r_{OS}$ the safety condition is achieved. The step of $\Delta_{r_{OS}}$ is 0.1 nm. The radius of the circle centered on $O_{1}$ remains constant at 5 nm.

While one aircraft flies a straight (linear) flight trajectory $B_{1}B_{2}$, the other aircraft flies a Dubins method flight trajectory $A_{1}A_{2}$, passing through 4 modes. The mathematical description of both aircraft flight trajectories, along with a more detailed graphical representation is provided below (Figure 3).

The proposed new model for aircraft conflict resolution analyzes the conflict between two aircraft by separating them in the horizontal plane according to two aspects (criteria):(1)Safety (i.e., ensuring a permanent horizontal separation of no less than 5 nm at all times).(2)The minimum (most rational) flight deviation from the flight plan (FPL) path.

### 2.1. Determination of Aircraft Movement on a Linear Flight Trajectory $B_{1}B_{2}$
$B_{1}B_{2}$


The initial (start) point $R_{B_{1}}$ coordinates vector of the second aircraft at time unit (t+Δt) is equal to:(1)$$R_{B_{1}}\left(t+\Delta t\right)=R\left(t\right)+\Delta tV_{w}(t)$$

The position vector of the second aircraft at time unit (*t* + Δ*t*) under the influence of wind uncertainty is equal to:
(2)$$R_{B_{1}}\left(t+\Delta t\right)=R_{L_{2}}(t)+\Delta t\left(V_{w}+e_{\tau ,L_{2}}V_{{L_{2}}_{0}}\right)$$
where, $e_{\tau ,L_{2}}$—the unit vector from the second aircraft’s position $R_{L_{2}}$(*t*) toward the final (simulation end) point $R_{B_{2}}$, which is calculated as follows:
(3)$$e_{\tau ,L_{2}}=\frac{R_{B_{2}}-R_{L_{2}}\left(t\right)}{\left|R_{B_{2}}-R_{L_{2}}\left(t\right)\right|\;}$$

The speed vector of the second aircraft at the time unit, i.e., moment (*t*+Δ*t*) under the influence of wind is equal to:(4)$$V_{L_{2}}\left(t+\Delta t\right)=e_{\tau ,L_{2}}\left(t+\Delta t\right)\left(V_{L2_{O}}+e_{\tau {,}_{L2}}^{T}\left(t+\Delta t\right)V_{W}\right)$$where,
$$e_{\tau {,}_{L2}}^{T}\left(t+\Delta t\right)=\frac{R_{B2}-R_{L2}\left(t+\Delta t\right)}{\left|R_{B2}-R_{L2}\left(t+\Delta t\right)\right|}$$

Since the the second aircraft’s follows a linear flight trajectory the first aircraft must modify its own linear flight trajectory *A*_1_*A*_2_, because it is slightly farther from the conflict point *O_s_* than second aircraft. Thus, the first aircraft should fly along the Dubins method trajectory via Mode 1–4 as shown in Figure 3. Therefore, the calculations for the first aircraft’s position and speed vectors are described below.

### 2.2. Determination of Aircraft Transitioning via 4 Modes on Flight Trajectory $A_{1}A_{2}$


As the the first aircraft flight trajectory is modified from a straight (linear) to ensure a minimum of 5 nm safe required horizontal separation between the two aircraft at all times, it follows a flight trajectory via four modes based on the Dubins method flight trajectory.

#### 2.2.1. Determination of Flight through Mode 1

The main length ($L_{A_{1},P_{1}})$ formula of mode 1 can be calculated as follows:(5)$$L_{A_{1,}P_{1}}=L_{A_{1,}O_{s}}-\sqrt{(r_{O_{1}}+{r_{O_{S}})}^{2}-r_{O_{1}}^{2}}$$

For more details, please refer to Formulas (5)–(8) in Dudoit et al. (2022) [50].

#### 2.2.2. Determination of Tangential Unit Vector

As displayed in Figure 4, the tangential unit vector $e_{\tau i}$ should be determined at each of the five-node discrete elements.

The tangential unit vector $e_{\tau i}$ as displayed in Figure 4 can be determined as follows:(6)$$e_{ni}=\frac{R_{L1}-R_{Os}}{\left|R_{L1}-R_{Os}\right|}$$
(7)$$e_{\tau i}=e_{ni}\times e_{3}=-e_{1}^{T}e_{ni}+e_{2}^{T}e_{ni}$$
where,i = 1…5.

The tangential unit vector $e_{\tau i}$ in the 5-node discrete element, as displayed in Figure 5 is equal to the following:(8)$$e_{\tau i}=\overset{5}{\underset{i=1}{\sum }}N_{i}(\xi )e_{\tau i}$$

If the aircraft position vector $R_{L1}\left(t\right)$ and $R_{Os}\left(t\right)$ vectors are known, the point P local coordinate $\xi_{p}$ is determined as follows:(9)$$\Phi =e_{\tau }^{T}\left(\xi_{p}\right)e_{np}=0$$

The algebraic non-linear Equation (8) is solved using the Newton-Rapson method:(10)$$\left(\frac{de_{\tau }{\left(\xi_{pk}\right)}^{T}e_{np}}{d\xi }\right)\Delta \xi_{pk}=-\Phi \left(e_{pk}\right)$$
where,
$$\xi_{\mathrm{pk}+1}{=\xi }_{\mathrm{pk}}+\Delta \xi_{\mathrm{pk}}$$

The determination of the aircraft coordinates and speed vectors at time unit (t+Δt), under the influence of uncertainties, is described below.

#### 2.2.3. Determination of Aircraft Coordinates and Speed Vectors at Time Unit (t+Δt)

According to the relationship between (8) and (10), the local coordinate $\xi_{p}\left(t\right)$ is determined at the element e and the tangential unit vector $e_{\tau p}$ is determined based on Formula (7).

The initial aircraft speed vector at time unit (t) is equal to the following:(11)$$V_{L1}\left(t\right)=e_{\tau p}\left(t\right)\left(V_{L1_{O}}+e_{\tau p}^{T}\left(t\right)V_{w}\right)+e_{np}\left(t\right)\left(e_{np}^{T}\left(t\right)V_{w}\right)R_{L1_{corr}}\left(t+\Delta t\right)=R_{L1}\left(t\right)+\Delta tV_{L1}\left(t\right)$$


The corrected/itemized aircraft position vector at time unit (t+Δt) is equal to the following:(12)$$R_{L1_{corr}}\left(t+\Delta t\right)=R_{L1}\left(t\right)+\Delta tV_{L1}\left(t\right)$$

The corrected/itemized aircraft speed at time unit (t+Δt) is equal to:(13)$$V_{L1_{corr}}\left(t+\Delta t\right)=e_{\tau }\left(t+\Delta t\right)\left(V_{L1_{O}}+e_{\tau }^{T}\left(t+\Delta t\right)V_{w}\right)+e_{n}\left(t+\Delta t\right)\left(e_{n}^{T}\left(t+\Delta t\right)V_{W}\right)$$

The final aircraft position vector at time unit (t+Δt) is equal to:(14)$$R_{L1}\left(t+\Delta t\right)=R_{L1}\left(t\right)+\frac{1}{2}\Delta t\left(V_{L1}\left(t\right)+V_{L1_{corr}}\left(t+\Delta t\right)\right)$$

Unit vectors $e_{\tau p}\left(t\right)$, $e_{np}\left(t\right)$, $e_{\tau p}\left(t+∆t\right)$, and $e_{\tau p}\left(t+∆t\right)$ are determined based on Formulas (6) and (7).

Vectors $R_{Os}\left(t+\Delta t\right)$ and $R_{Os2}\left(t+\Delta t\right)$ are determined as follows:(15)$$R_{Os}\left(t+\Delta t\right)=R_{Os}\left(t\right)+\Delta tV_{W}(t)$$
(16)$$R_{Os2}\left(t+\Delta t\right)=R_{Os2}\left(t\right)+\Delta tV_{W}(t)$$

The first aircraft terminates its flight via the circle with center $O_{s}$ when the local coordinate $\xi_{p}$ at point P, located on the circle, becomes greater than the local coordinate of a point $\xi_{\tau }$, which is on the circle to which the tangential line from the simulation endpoint $A_{2}$ is drawn (Figure 6).

The local coordinate $\xi_{\tau }$ is determined solving the non-linear algebraic equation:(17)$$\Phi \left(\xi_{\tau }\right)=R_{AR}^{T}\frac{dR(\xi_{p})}{d\xi }-\left|R_{AR}\right|\left|\frac{dR(\xi_{\tau })}{d\xi }\right|=0$$where, $$R_{AR}=R_{A2}-R\left(\xi_{\tau }\right)$$

Formula (17) is solved applying the Newton-Rapson method: (18)$$\frac{d\Phi \left(\xi_{k}\right)}{d\xi }\Delta \xi_{k}=-\Phi (\xi_{k})$$where, k- iteration number.



$$\xi_{k+1}=\xi_{k}+\Delta \xi_{k}$$



Local coordinate $\xi_{p}$ is determined by solving Formula (10).

The results obtained from the proposed mathematical model investigation are pre-sented in the following section.

### 2.3. Numerical Investigation

The model was programmed using Maple code, consisting of 100 lines with 10 input parameters and 5 output parameters.

An integration step of 0.01 [s] was used for the calculations on the 11th Gen Intel(R) Core (TM) i5-1135G7 2.42 GHz RAM 8GB and took 3 h, involving 80 steps × 4 cases = 320 iterations.

## 3. Results

Using the proposed mathematical model for aircraft conflict resolution trajectories under uncertainties (like wind), the following results were obtained: optimal main circle radii ($r_{Os}$), which have a deciding consequence on the minimal distance (${\Delta L}_{min}$) between the two aircraft flight trajectories, and variations (values) in two aircraft speeds $V_{1}$ and $V_{2}$.

### Optimal Radii and Minimal Distance Analysis

For the windless case ($V_{w}$ = 0 kts), when the initial angle (α) between the two aircraft flight trajectories is 35°; the main circle radii $r_{Os}$ (see Figure 3) amounts to 8.8 nm, and minimum distance ${\Delta L}_{min}$ between the two flight trajectories is 5.0 nm. When the initial angle (α) is 45°, for the same main circle radii $r_{Os}$ of 8.8 nm, the minimum distance ${\Delta L}_{min}$ between the two flight trajectories is 2.71 nm. For an initial angle (α) of 55°, with the same main circle radii $r_{Os}$ of 8.8 nm, the minimum distance ${\Delta L}_{min}$ between the two flight trajectories is 0.74 nm. However, the three-cases (35°, 45° and 55°) analysis were expanded to include the investigation of four additional constant wind directions ($\alpha_{w}=$ 0°, 90°, 180° and 270°) and wind speeds ($V_{w}$ = 20 kts) to ensure both a safe and rational resolution of a conflict between the two aircraft. The received values were presented hereinunder in Table 1 and with some of them graphically illustrated in Figure 7.

The adapted results of the minimum distance ${\Delta L}_{min\_i}$ between two aircraft flight trajectories for one of the three initial angle cases (α=35°,45°,55°) clearly showed that, in a windless situation ($V_{w}$ = 0 kts) with an initial angle of 35° case; the distance ${\Delta L}_{min\_i}$ between the two flight trajectories was ensured and amounted at 5.0 nm. For a wind direction of 0° and a speed of 20 kts, the distance ${\Delta L}_{min\_i}$ decreased to 4.17 nm. When the wind direction was 90° with the speed of 20 kts, the minimum distance amounted to 5.29 nm. For wind directions of 180° and 270° at a speed of 20 kts, the distances amounted to 5.31 nm and 4.26 nm, respectively. This explains that wind can either impact the assurance of a safe distance (as in the cases of wind direction of 90° or 180° and speed of 20 kts), or reduce it (as in the cases of wind direction of 0° or 270° and speed of 20 kts).

The graphically illustrated minimum distances for the initial calculation model results are shown in Figure 7 below.

Based on the results (shown in Table 1 and Figure 7), it was identified that violations of the safe 5 nm minimum required horizontal distance between the two aircraft flight trajectories occurred when the radii $r_{Os}$ remained fixed at 8.8 nm. Therefore, the primary goal is to ensure a minimum safe horizontal distance of no less than 5 nm at all times, and consequently maintain rational flight trajectories for both aircraft. Therefore, the radii $r_{Os}$ should be recalculated for the three analyzed three cases with different wind direction values.

The following recalculated results were obtained to ensure the safe distance between the two aircraft flight trajectories using the proposed mathematical model for the aircraft conflict resolution trajectories under uncertainties (like wind). These recalculated results are illustrated together with the initial modelling results for comparison.

For the recalculated windless case ($V_{w}$ = 0 kts), when the initial angle (α) between the two aircraft flight trajectories was 35°, the main circle radii $r_{Os}$ (see Table 2) was 9.42 nm, resulting in a minimum distance ${\Delta L}_{min\_r2}$ of 5.89 nm between the two flight trajectories. When the initial angle was 45°, the main circle radii $r_{Os}$ (see Table 2) amounted to 13.20 nm and the ${\Delta L}_{min\_r2}$ between two flight trajectories amounted to 5.21 nm. For an initial angle of 55°, the main circle radii $r_{Os}$ (see Table 2) was 12.40 nm, and the ${\Delta L}_{min\_r2}$ between the two flight trajectories amounted to 7.46 nm.

However, the recalculated analysis for the three-cases (35°, 45° and 55°) was expanded to include an investigation of four additional constant wind directions ($\alpha_{w}=$ 0°, 90°, 180° and 270°) and wind speeds ($V_{w}$ = 20 kts) to ensure both a safe and rational resolution of a conflict between the two aircraft. The received recalculated values obtained are presented in Table 2 below and are graphically illustrated with initial values in Figure 8.

The adapted recalculated results of the minimum distance ${\Delta L}_{min}$ between two aircraft flight trajectories for each of the three cases clearly showed the following: for the 35° case in the headwind (180°) situation, the greatest distance between the two flight trajectories was ensured, amounting to 6.66 nm; for the 45° case in same headwind situation, the greatest distance between two flight trajectories was ensured, amounting to 7.45 nm; and for the 55° case in the same headwind situation, the safe distance between two flight trajectories was ensured, amounting to 8.28 nm. This shows that the headwind situation ensures the greatest distance values between the two aircraft flight trajectories across the analyzed configurations. However, in the 35° case with a crosswind (270°) situation, a smaller distance between the two flight trajectories was observed, amounting to 5.25 nm. For the 45° case in a windless situation, the smallest distance between the two flight trajectories was ensured, amounting to 5.21 nm, and for the 55° case with a tailwind (0°) situation, the smallest distance between the two flight trajectories was ensured, amounting to 6.63 nm. It was determined that, for the 35° case, the ratio between the greatest and the smallest distance values was 21%, for the 45° case, the ratio was 30%; and for the 55° case the ratio was 20%. This variation may be explained by all parameter combinations such as wind direction, wind speed, aircraft speed and the influence of the initial angle between the two aircraft flight trajectories.

Based on the results above (Table 2 and Figure 8), it is evident that after recalculating the radii $r_{os}$, the minimal safe required horizontal distance of no less than 5 nm was ensured. Thus, allowing for rational flight trajectories in respect to the flight planned (FPL) trajectories for both aircraft.

## 4. Validation of the Results

As the proposed mathematical model ensures both safe and rational flight trajectories, the results of the study on the location of two aircraft conflicts under uncertainty conditions were compared with the results from the Romero et al. (2020) [4], which analyzed the effect of wind on two aircraft flight trajectories conflict, as presented in Figure 9. Differences in initial conditions, simulation parameters, and aircraft speeds can explain the difference in flight trajectory distribution. It can be argued that the set of variable parameters applied in this research allows for a more accurate realization of flight trajectory distribution.

After comparing the results of the proposed mathematical model with those of Romero et al. (2020) [4], as presented in Figure 9, it was found that the proposed mathematical model allowed flight trajectories to be close enough while still meeting safety criteria (maintaining a minimum distance of 5 nm at all times). Additionally, the model ensured smaller deviations from the FPL trajectory, resulting in a more rational use of airspace. This could allow for a higher number of aircraft flying within it.

## 5. Conclusions

As air traffic intensity increases, it becomes important to search for new means of aircraft conflict resolution to ensure the required level of safety and rational flight trajectories. A literature review showed that stochastic uncertainties significantly impact air traffic controllers’ workload, together with pressure from airlines to reduce costs by flying via straighter (more direct) trajectories. This explains that it is worth offering a new aircraft conflict resolution model to ensure a safe and rational (economical) solution to managing aircraft conflict situations. Such a solution could be implemented using Dubin’s method flight trajectories in a dynamic conflict resolution.Using the proposed mathematical model, the adapted results obtained for the minimum distance Δ*L* between two aircraft flight trajectories for each investigated case under certain uncertainties clearly showed that for the 35° case, the ratio between the greatest and the smallest distance values amounted to 21%; for the 45° case the ratio was 30%; and for the 55° case the ratio was 20%. This might be explained by combining all parameters, such as wind direction, wind speed, aircraft speed, and the significant impact of the initial angle between the aircraft flight trajectoriesUsing the proposed mathematical model, the adapted results obtained for the random speed of two aircraft in each investigated case under certain uncertainties clearly showed that the proposed model ensured safety criteria (no less than 5 nm at all times) and achieved a rational smaller deviation with higher density compared to the model proposed by Romero et al. (2020) [4]. Consequently, this had a positive impact on the more rational use of airspace and allowed for a higher number of aircraft flying within it.As the proposed method could be applied when wind direction and speed are inconstant (stochastic) during simulation, further studies could be carried out with this proposed mathematical model in cases where wind direction and speed are random (stochastic) throughout the simulation. Additionally, the proposed study could be carried out using unmanned aircraft systems.

## Figures and Tables

**Figure 1 sensors-24-05877-f001:**
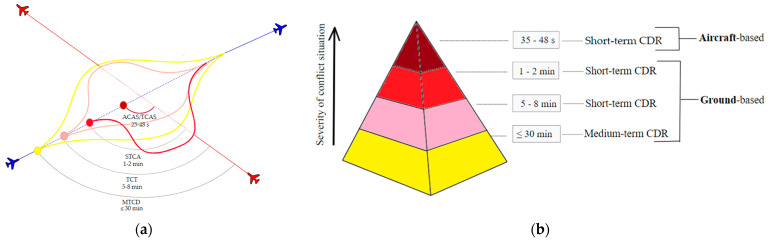
(**a**) A simplified representation of a two-aircraft conflict solution; (**b**) ground-based and aircraft-based CDR (based on: EUROCONTROL, 2017 [11]).

**Figure 2 sensors-24-05877-f002:**
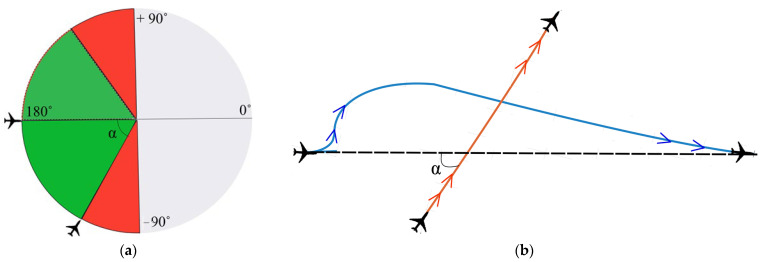
(**a**) Numerical model of suitable angles between flight trajectories: green—suitable for conflict investigation; red—not applicable; gray—out of scope due to typical aircraft operating procedures. (**b**) Graphical representation a conflict situation between the flight paths of two aircraft.

**Figure 3 sensors-24-05877-f003:**
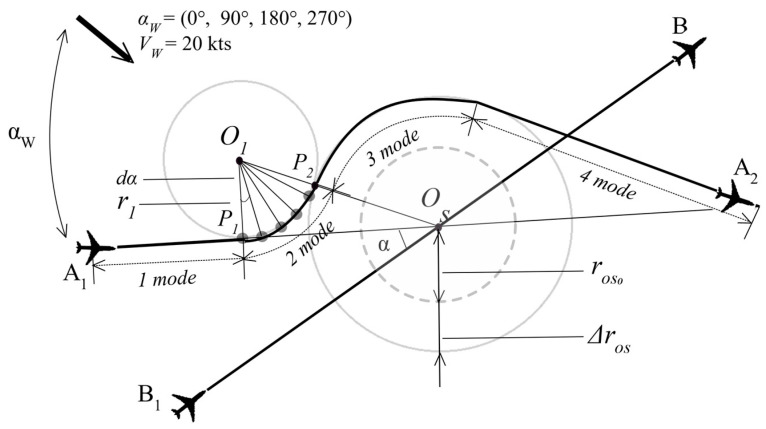
Configuration of the analytical investigation model with different wind angles ($\alpha_{W}$ = 0°, 90°, 180°, 270°) and different initial angles between flight trajectories (α = 35°, 45°, 55°).

**Figure 4 sensors-24-05877-f004:**
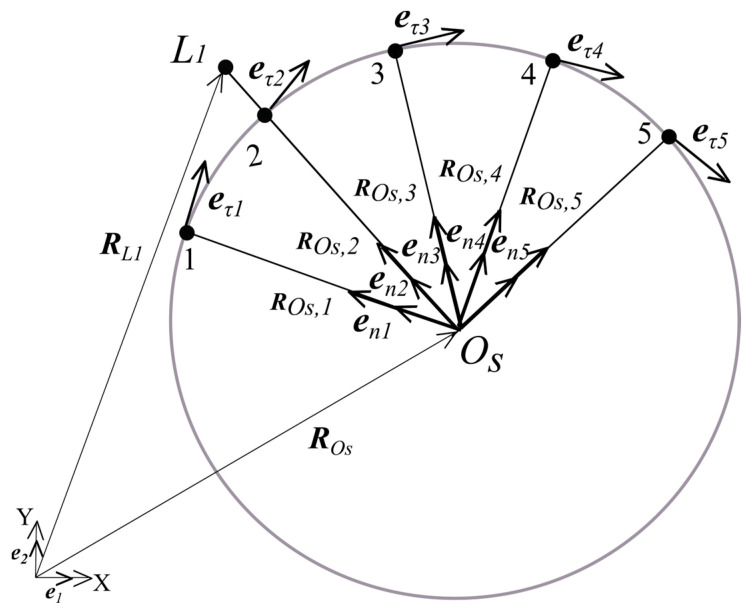
One discrete element with 5 nodes of the tangential unit $e_{\tau i}$ vector.

**Figure 5 sensors-24-05877-f005:**
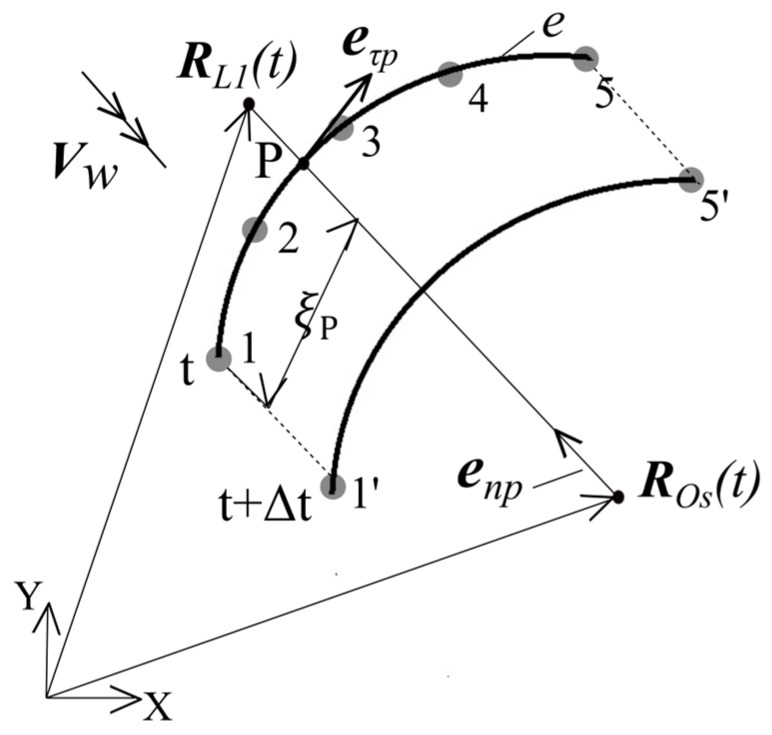
Point P local coordinate $\xi_{p}$.

**Figure 6 sensors-24-05877-f006:**
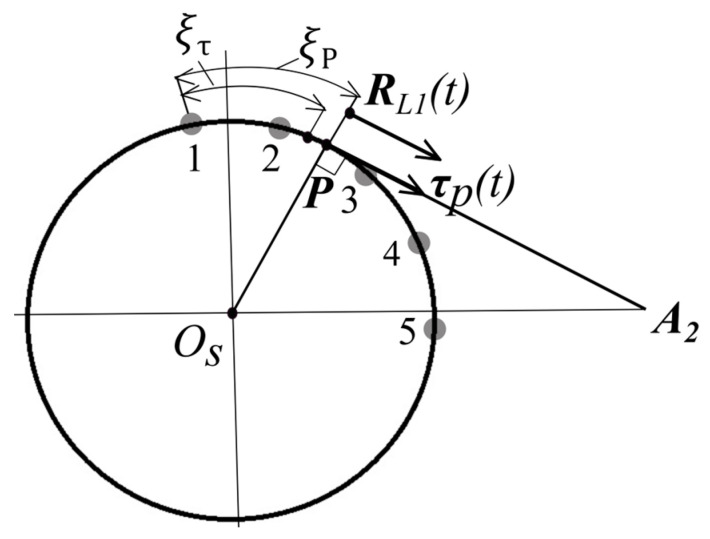
Termination of flight via the arc and the start of flight via tangent.

**Figure 7 sensors-24-05877-f007:**
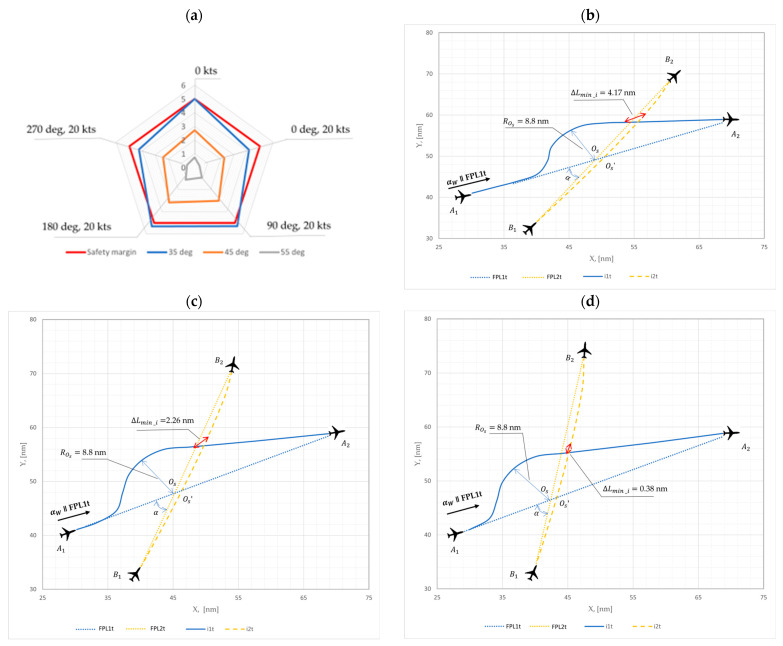
Aircraft conflict situation initial modelling of the minimal horizontal distance between two aircraft flight trajectories, when the initial angle (α) amounted to: (**a**) overall case, (**b**) case 35°, (**c**) case 45°, and (**d**) case 55°.

**Figure 8 sensors-24-05877-f008:**
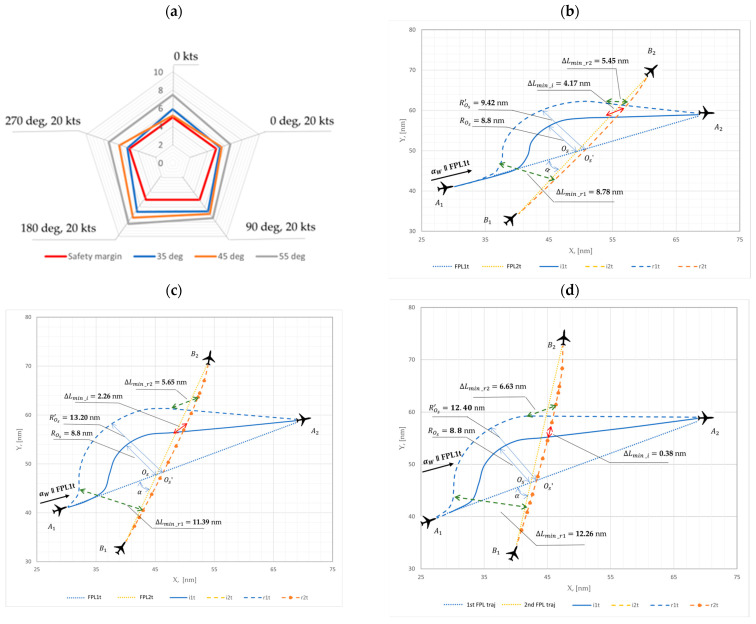
Aircraft conflict situation initial and adapted recalculated modelling of the minimal horizontal distance between two aircraft flight trajectories, with the initial angle amounting to: (**a**) overall case (**b**) case 35°; (**c**) case 45° and (**d**) case 55°.

**Figure 9 sensors-24-05877-f009:**
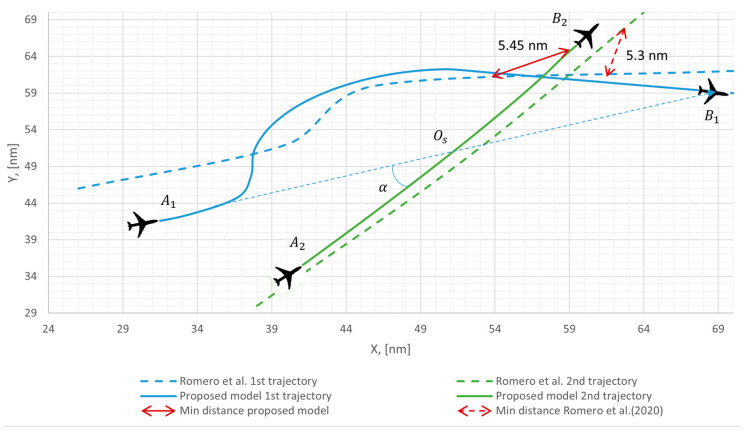
Comparison of the obtained results with Romero et al. (2020) [4], when $\alpha_{w}$ = 0° and $V_{w}$ = 20 kts.

**Table 1 sensors-24-05877-t001:** Radii $r_{Os}$ and minimum distance ${\Delta L}_{min}$ values for the initial angle (α) in three analyzed cases.

**Initial Angle between Trajectories** (α)	**Wind** $V_{w}0$ = 0 kts		**Wind** $\alpha_{w}$ = 0° $V_{w}$ = 20 kts		**Wind** $\alpha_{w}$ = 90° $V_{w}$ = 20 kts		**Wind** $\alpha_{w}$ = 180° $V_{w}$ = 20 kts		**Wind** $\alpha_{w}$ = 270° $V_{w}$ = 20 kts	
	$r_{os}$, [nm]	${\Delta L}_{min},\left[nm\right]$	$r_{os},$ [nm]	${\Delta L}_{min},\left[nm\right]$	$r_{os,}$ [nm]	${\Delta L}_{min},\left[nm\right]$	$r_{os},$ [nm]	${\Delta L}_{min},\left[nm\right]$	$r_{os},$ [nm]	${\Delta L}_{min},\left[nm\right]$
35°	8.8	5.0	8.8	4.17	8.8	5.29	8.8	5.31	8.8	4.26
45°		2.71		2.26		3.0		3.15		2.42
55°		0.74		0.38		0.91		1.10		0.58

**Table 2 sensors-24-05877-t002:** Recalculated radii $r_{Os}$ and minimum distance ${\Delta L}_{min}$ values for the three analyzed cases.

**Initial Angle between Trajectories** (α)	**Wind** $V_{w}$ = 0 kts		**Wind** $\alpha_{w}$ = 0° $V_{w}$ = 20 kts		**Wind** $\alpha_{w}$ = 90° $V_{w}$ = 20 kts		**Wind** $\alpha_{w}$ = 180° $V_{w}$ = 20 kts		**Wind** $\alpha_{w}$ = 270° $V_{w}$ = 20 kts	
	$r_{Os}$, [nm]	${\Delta L}_{min},\left[nm\right]$	$r_{Os}$, [nm]	${\Delta L}_{min},\left[nm\right]$	$r_{Os}$, [nm]	${\Delta L}_{min},\left[nm\right]$	$r_{Os}$, [nm]	${\Delta L}_{min},\left[nm\right]$	$r_{Os}$, [nm]	${\Delta L}_{min},\left[nm\right]$
35°	9.42	5.89	9.42	5.45	9.42	6.58	9.42	6.66	9.42	5.25
45°	13.20	5.21	13.20	5.65	13.20	6.94	13.20	7.45	13.20	6.19
55°	12.40	7.46	12.40	6.63	12.40	7.49	12.40	8.28	12.40	7.38

## Data Availability

The data presented in this study are available on request from the corresponding author.

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
