# Peer review of "Investigation of Aircraft Conflict Resolution Trajectories under Uncertainties"

_sensors, 2024, doi:10.3390/s24185877_

Round 1

Reviewer 1 Report

Comments and Suggestions for Authors

This paper points out that in order to cut costs and reduce the workload of air traffic controllers, airlines will make aircraft adopt more direct flight paths, and wind uncertainty is an important factor affecting the flight path of aircraft, which will have a direct impact on the resolution of aircraft conflict situations. Therefore, based on the current CDR model, this paper focuses on taking wind uncertainty into account. A new CDR model for dynamic conflict resolution using Dubins method is proposed. The model takes into account different initial angles of aircraft flight trajectories and, on the premise of minimizing deviation from the original flight plan, ensures a minimum horizontal interval of 5 kilometers between flight paths by adjusting the flight trajectory radius, so as to solve the flight conflict problem safely and efficiently. The model takes into account the influence of wind factors, which is conducive to reasonable planning of navigation trajectory, and can improve safety and work efficiency. But before this article is published, I think the following aspects need to be optimized.

1.The DSS functions currently used in CDR, such as MTCD, TCT, STCA and ACAS/TCAS, are best expressed more intuitively and clearly in the form of flow charts.

2.In the fifth paragraph of the introduction, there should be a brief summary of the content of the random method used for CDR in literature [14-16].

3.In Section 2.1, is it too idealistic to consider only the collision between two aircraft in the simulation hypothesis? Can we increase the sample size?

4.In Section 3.1, only three initial angles of 35°, 45° and 55° are considered in the analysis of optimal radius and minimum distance. Should it be extended to comprehensively analyze the optimal solution under different circumstances?

5.It is suggested to further explore the effect of different wind speed and direction on the minimum horizontal distance.

6.In addition to the wind factor, the model should also consider other factors, such as other weather conditions, aircraft performance, human intervention, etc., to enhance the universality of the model.

7.In Section 4, the comparison between the experimental results obtained by the model and the real airline flight data should be added to check the gaps.

8.In Figure 8, wind speed, wind direction and navigation position coordinates should be marked more clearly, so as to facilitate readers to understand the simulation process of the model more intuitively.

9.As for the references, we should pay attention to the timeliness, control the number of classical references, and refer to the relevant literatures in the past two years to ensure the prospective research.

Comments on the Quality of English Language

Fine

Author Response

Dear, Reviewer,

Thank you very much for Your valuable remarks and recommendations. Please see the attached Word file (Review Report Form1.docx) with our responses.

Kind regards,

A.Dudoit

Reviewer 2 Report

Comments and Suggestions for Authors

This article has some innovation, but I have the following comments:

1.     The simulation figures are not clear enough, as shown in Figures 7 and 8.

2.     There should be a technical summary cited in section 2.2.1, rather than simply providing a reference article.

3.     This article proposes a mathematical model of flight trajectory. How to verify whether it conforms to the flight dynamics model during actual flight?

4.     Is there a more convenient flight trajectory encounter model in the three-dimensional direction?

Comments on the Quality of English Language

n/a

Author Response

Dear Reviewer,

Thank you very much for Your remarks and recommendations. Please see our responses in the attached Word File (see: Review Report Form 2.docx).

Kind regards,

A.Dudoit

Round 2

Reviewer 1 Report

Comments and Suggestions for Authors

Accept in present form

Comments on the Quality of English Language

Fine

Author Response

Dear Reviewer, 

thank you for your remark. We have revised the paper introduction according to your remarks (CDR methods, references and comprehensive English language revision). We hope it looks better and we are looking forward for your response.

Kind regards,

Anrieta Dudoit
